# Heterologous HSPC Transplantation Rescues Neuroinflammation and Ameliorates Peripheral Manifestations in the Mouse Model of Lysosomal Transmembrane Enzyme Deficiency, MPS IIIC

**DOI:** 10.3390/cells13100877

**Published:** 2024-05-20

**Authors:** Xuefang Pan, Antoine Caillon, Shuxian Fan, Shaukat Khan, Shunji Tomatsu, Alexey V. Pshezhetsky

**Affiliations:** 1Department of Pediatrics and Centre Hospitalier Universitaire Sainte-Justine Research Centre, University of Montreal, Montreal, QC H3T 1C5, Canada; xuefang.pan.hsj@ssss.gouv.qc.ca (X.P.); antoine.caillon.etud@gmail.com (A.C.); shuxian.fan@mail.mcgill.ca (S.F.); 2Department of Anatomy and Cell Biology, McGill University, Montreal, QC H3A 0C7, Canada; 3Nemours/Alfred I. duPont Hospital for Children, Wilmington, DE 19803, USA; shaukat.khan@nemours.org (S.K.); shunji.tomatsu@nemours.org (S.T.)

**Keywords:** mucopolysaccharidosis, Sanfilippo disease, neuroinflammation, heparan sulfate, HSPC, allogenic HSPC transplantation, microglia

## Abstract

Mucopolysaccharidosis III type C (MPS IIIC) is an untreatable neuropathic lysosomal storage disease caused by a genetic deficiency of the lysosomal N-acetyltransferase, HGSNAT, catalyzing a transmembrane acetylation of heparan sulfate. HGSNAT is a transmembrane enzyme incapable of free diffusion between the cells or their cross-correction, which limits development of therapies based on enzyme replacement and gene correction. Since our previous work identified neuroinflammation as a hallmark of the CNS pathology in MPS IIIC, we tested whether it can be corrected by replacement of activated brain microglia with neuroprotective macrophages/microglia derived from a heterologous HSPC transplant. Eight-week-old MPS IIIC (*Hgsnat^P304L^*) mice were transplanted with HSPC from congenic wild type mice after myeloablation with Busulfan and studied using behavior test battery, starting from the age of 6 months. At the age of ~8 months, mice were sacrificed to study pathological changes in the brain, heparan sulfate storage, and other biomarkers of the disease. We found that the treatment corrected several behavior deficits including hyperactivity and reduction in socialization, but not memory decline. It also improved several features of CNS pathology such as microastroglyosis, expression of pro-inflammatory cytokine IL-1β, and accumulation of misfolded amyloid aggregates in cortical neurons. At the periphery, the treatment delayed development of terminal urinary retention, potentially increasing longevity, and reduced blood levels of heparan sulfate. However, we did not observe correction of lysosomal storage phenotype in neurons and heparan sulfate brain levels. Together, our results demonstrate that neuroinflammation in a neurological lysosomal storage disease, caused by defects in a transmembrane enzyme, can be effectively ameliorated by replacement of microglia bearing the genetic defect with cells from a normal healthy donor. They also suggest that heterologous HSPC transplant, if used together with other methods, such as chaperone therapy or substrate reduction therapy, may constitute an effective combination therapy for MPS IIIC and other disorders with a similar etiology.

## 1. Introduction 

Lysosomal storage disorders (LSDs) are conditions caused by the accumulation (storage) of biological macromolecules due to genetic defects in lysosomal catabolism. More than 70 LSDs are currently described that, together, have frequency of 1:4000 live births [1]. About 30% of children with LSDs are affected by mucopolysaccharidoses (MPS) caused by the impaired lysosomal degradation of glycosaminoglycans (GAGs). Neurological MPS (MPS I, MPS II, MPS IIIA, MPS IIIB, MPS IIIC, MPS IIID, and MPS VII) are characterized by the accumulation of a specific GAG, heparan sulphate (HS) (reviewed in ref. [2]). HS, primarily stored in the phagocytic microglia and astrocytes, triggers their activation and release of pro-inflammatory cytokines and chemokines [3], which can further stimulate the upregulation of adhesion molecules on brain endothelial cells, resulting in enhanced trans-endothelial migration of monocytes from the blood into perivascular regions. These enzyme-deficient cells migrate into the parenchyma, leading to further damage and cell death (reviewed in ref. [2]). In particular, we have previously shown that in the MPS IIIC mouse model *Hgsnat-Geo*, a functional knockout (KO) of the *Hgsnat* locus in mice, storage of HS in microglia leading to their activation was the initial pathological event in the brain, present already at 2 months and gradually aggravating with age [4]. Microglia activation resulted in the release of pro-inflammatory cytokines, including IL-1β, that cause mitochondrial damage [5,6,7]. Together with accumulation of misfolded toxic protein aggregates, this eventually leads to neuronal death, dominant in all brain regions. More recently, we confirmed that a similar pathophysiological mechanism also manifests in a knock-in MPS IIIC mouse strain *Hgsnat^P304L^*, that expresses a mutant misfolded HGSNAT protein [8]. This study defined CD68-positive microglia and GFAP-positive activated astrocytes as the most robust biomarkers of the disease progression [4,8]. We also confirmed that similar biomarkers are present in post mortem cortical tissues of human Sanfilippo patients, which suggested that the described pathophysiological mechanism is common for mice and humans [9].

Non-neurological LSDs including MPS I-Scheie, MPS IVA, and MPS VI are partially treatable by enzyme replacement therapy (ERT), but this approach is ineffective for the neurological forms due to the inability of recombinant enzymes to cross the blood–brain barrier (BBB). In MPS III type C, caused by genetic deficiency of the lysosomal membrane enzyme, heparan sulfate acetyl-CoA: α-glucosaminide N-acetyltransferase (HGSNAT), an additional constraint is the absence of cross-correction between the cells. Delivering a replacement enzyme intrathecally or intraventricularly, thus bypassing the BBB [10], is not possible for membrane enzymes and proteins such as HGSNAT and is difficult to implement clinically due to the invasive nature of the procedure.

Hematopoietic stem and progenitor cell (HSPC) transplant is the clinically recommended therapy for severe neuropathic forms of MPS I, where the missing enzyme is soluble and can be effectively secreted by donor cells [11]. The treatment restores catabolism in HSPC which, in combination with cross-correction of other cell types through secretion of the enzyme, reduces storage in the tissues of peripheral organs. Importantly, the donor-derived monocytes also cross the BBB and differentiate into brain macrophages/microglial cells, thus ameliorating the neuropathology through cross-correction of neurons [12,13]. It is not clear whether this methodology shows efficacy in MPS IIIC where the defective enzyme is a transmembrane protein; however, the wild type (WT) brain macrophages/microglia derived from the transplanted HSPC could potentially mitigate neuroinflammatory response.

In the current study, we show that neuroinflammation in the brains of *Hgsnat^P304L^* mice is indeed effectively corrected by allogenic transplantation of HSPC from WT mice, despite the absence of a rescue of HGSNAT activity or HS storage in the brain. Transplantation conducted in 2-month-old *Hgsnat^P304L^* mice also prevented the development of urinary retention, thus extending the life span of animals beyond the age of 8 months. Together, our results demonstrate that neuroinflammation is the crucial underlying pathological mechanism in MPS IIIC that can be effectively treated by replacement of microglia bearing the genetic defect with cells from a normal healthy donor. Still, behavioral deficits and pathological changes in the neurons showed only partial improvement, suggesting that HSPC transplantation needs to be combined with other interventions for the development of an effective therapy.

## 2. Materials and Methods

### 2.1. Animals

The knock-in mouse model of MPS IIIC, *Hgsnat^P304L^*, expressing the HGSNAT enzyme with human missense mutation P304L, on a C57BL/6J genetic background, has been previously described [8]. C57BL/6 congenic mouse strain B6.SJL-Ptprca Pepcb/BoyJ, carrying the differential *Ptprc* pan leukocyte marker CD45.1 (Ly5.1), was purchased from The Jackson Laboratory (Stock No. 033076). The animals were housed in the Animal facilities of CHU Ste-Justine, following the guidelines of the Canadian Council on Animal Care (CCAC). The animals were kept in an enriched environment with a 12 h light/dark cycle, fixed temperature, humidity, and continuous access to water and food.

Equal cohorts of male and female mice were studied separately for each experiment, and statistical methods were used to test whether the progression of the disease, levels of biomarkers, or response to therapy were different for male and female animals. Since differences between sexes were not detected, the data for male and female mice were pooled together.

### 2.2. Isolation and Transplantation of HSPC

Tibia, femur, and iliac bones were dissected and washed with 70% (*v*/*v*) ethanol, followed by 3 washes with PBS containing 1% (*w*/*v*) penicillin and streptomycin. Under sterile conditions, bone marrow was flushed with ice-cold SCGM media using 25G syringe, and cells were dissociated with a glass pipette, strained through 70 μm nylon mesh and centrifuged at 380× *g* for 10 min at 4 °C. The pellet was resuspended in 10 mL of red blood cells (RBC) lysis buffer (155 mM NH_4_Cl, 12 mM NaHCO_3_, 0.1 mM EDTA) and incubated for 1 min followed by addition of 20 mL of ice-cold SCGM media. To isolate HSPC, cells were harvested by centrifugation at 380× *g* for 10 min at 4 °C and resuspended in EasySepT Buffer (Stem Cell Technologies, Toronto, ON, Canada). HSPCs were purified by negative selection using the EasySep Mouse Hematopoietic Progenitor Cell Isolation Kit (Stem Cell Technologies), according to the manufacturer’s instructions, and maintained in culture in 10 mL of SCGM media supplemented with 10 ng/mL rmIL-6, 100 ng/mL rmFlt-3L, 100 ng/mL rmTPO, and 100 ng/mL rmSCF at 37 °C with 5% CO_2_. After 24 h, cells were resuspended in PBS, counted, and injected in the tail vein of *Hgsnat^P304L^* mice (2.4–2.8 × 10^6^ cells per mouse in 100 μL of PBS). Prior to HSPC transplantation, the recipient mice were treated for 5 consecutive days with busulfan (IP, 25 mg/kg BW/day). During the myeloablation and for 15 days following the transplantation, mice were housed in pathogen-free conditions, fed with food sterilized by irradiation, and treated with Baytril^®^ (enrofloxacin, 50 mg/mL in drinking water).

### 2.3. Flow Cytometry Analysis of Blood Leucocytes

Blood from the mouse mandibular vein (50 μL–100 μL) was collected in EDTA-coated capillary and was stained for 15 min in the dark at room temperature with APC-labeled anti-mouse CD45.1 antibody (Biolegend, San Diego, CA, USA, 1:20) and PE-labeled anti-mouse CD45.2 antibody (Biolegend, 1:20). Erythrocytes were lysed with RBC lysis buffer for 5 min, as described above, and white cells were collected by centrifugation for 5 min at 380× *g* and washed twice with PBS containing 1% FBS. Fluorescence compensation was performed with anti-mouse Ig, κ/Negative Control (FBS) Compensation Particles Set (BDTM CompBeads) according to manufacturer’s instructions, and flow cytometry analysis was performed using FACSCanto II instrument (BD Biosciences, Mississauga, ON, Canada).

### 2.4. Enzyme Activity Assays

Enzymatic activities of HGSNAT and total β-hexosaminidase, were assayed essentially as previously described [14,15,16]. Tissues extracted from mice were snap-frozen in liquid nitrogen before storage at −80 °C. Samples of 50 mg were homogenized in 250 µL of H_2_O using a sonic homogenizer (Artek Systems Corporation, Bothell, WA, USA). For HGSNAT assays, 20 µL aliquots of the homogenates were combined with 10 µL of McIlvain Buffer (pH 5.5), 10 µL of 3 mM 4-methylumbelliferyl-β-D-glucosaminide (Biosynth, Compton, UK), and 10 µL of 5 mM acetyl-coenzyme A in H_2_O. The reaction was carried out for 3 h at 37 °C, stopped with 200 µL of 0.4 M glycine buffer (pH 10.4), and fluorescence was measured using a ClarioStar plate reader (BMG Labtech, Mornington, Australia). Blank samples were incubated without homogenates which were added after the glycine buffer.

The activity of β-hexosaminidase was measured by combining 2.5 μL of 10-times diluted homogenate (~2.5 ng of protein) with 15 μL of 0.1 M sodium acetate buffer (pH 4.2), and 12.5 µL of 3 mM 4-methylumbelliferyl N-acetyl-β-D-glucosaminide (Sigma-Aldrich, Saint Louis, MO, USA) followed by incubation for 15 min at 37 °C. The reaction was stopped with 200 µL of 0.4 M glycine buffer (pH 10.4), and fluorescence was measured as above.

### 2.5. Real-Time qPCR

RNA was isolated from snap-frozen brain tissues using the TRIzol reagent (Invitrogen, Waltham, MA, USA) and reverse-transcribed using the Quantitect Reverse Transcription Kit (Qiagen # 205311, Hilden, Germany) according to the manufacturer’s protocol. qPCR was performed using a LightCycler^®^ 96 Instrument (Roche, Basel, Switzerland) and SsoFast™ EvaGreen^®^ Supermix with Low ROX (Bio RAD #1725211, Hercules, CA, USA) according to the manufacturer’s protocol. RLP32 mRNA was used as a reference control. The forward (F) and reverse (R) primers used for qPCR are shown in the Table 1.

### 2.6. Behavioral Analysis

Short-term recognition memory was assessed by novel object recognition test [17,18]. Mice were placed individually in a 44 × 45 × 40 cm (length × width × height) testing chamber with white Plexiglas walls for 10 min habituation period and returned to their home cage. The next day, mice were placed for 10 min in the same testing chamber containing two identical objects (red plastic towers, 3 × 1.5 × 4.5 cm), returned to the home cages, and 1 h later, placed back for 10 min into the testing chamber containing one of the original objects and a novel object (a blue plastic base, 4.5 × 4.5 × 2 cm). After each mouse, the arena and the objects were cleaned with 70% ethanol to avoid olfactory cue bias. The discrimination index (DI) was calculated as the difference in the exploration time between the novel and old object divided by the total exploration time. A preference for the novel object was defined as a DI significantly higher than 0 [19]. Mice who showed a side preference, noted as a DI of ±0.20 during familiarization period, and those who had a total exploration time lower than 3 s, were excluded from the analysis.

The open-field test was performed as previously described [20]. Briefly, mice were habituated in the experimental room for 30 min before the commencement of the test. Then, each mouse was gently placed in the center of the open-field arena and allowed to explore for 20 min. The mouse was removed and transferred to its home cage after the test, and the arena was cleaned with 70% ethanol before the next test. The behavioral activity was analyzed using the Smart video tracking software (v3.0, Panlab Harvard Apparatus); the total distance traveled, and percent of time spent in the center zone were measured for hyperactivity and anxiety assessment, respectively.

Contextual fear conditioning used to analyze the ability of mice to associate an environment (context) with a fear-inducing stimulus was performed as previously described [21]. Briefly, one day prior to fear conditioning (Day 0), mice were allowed to freely explore the context for habituation for 3 min. On Day 1, mice were trained using a fear-inducing stimulus (5.5 min in total, 3 random 0.6 mA electric shocks, 2 sec each time). On Day 2 and Day 7, conditioned mice were placed in the context chamber for 5 min. The freezing time was calculated based on the video records.

Social interaction behavior was studied using Crawley’s sociability and preference for social novelty test, as described in [22]. Briefly, mice were allowed to freely explore a rectangular, three-chamber box with each chamber measuring 19 × 45 cm and the dividing walls made from clear Plexiglas, with an open middle section and free access to each chamber. Side chambers contained two identical wire cup containers, one empty and one holding a naïve/unfamiliar mouse. All chambers were cleaned with 70% ethanol after each trial. Mice were habituated for 5 min, after which the walls between the compartments were removed to allow free access for the subject mouse to explore each of the three chambers for the duration of 10 min. The time spent in each chamber and the time spent interacting with wire containers were calculated based on the video records.

### 2.7. Immunohistochemistry

Mouse brains were collected from animals, perfused with 4% PFA in PBS and post-fixed in 4% PFA in PBS overnight. Brains were cryopreserved in 30% sucrose for 2 days at 4 °C, embedded in Tissue-Tek^®^ OCT Compound, cut in 40 µm thick sections and stored in cryopreservation buffer (0.05 M sodium phosphate buffer, pH 7.4, 15% sucrose, 40% ethylene glycol) at −20 °C pending immunohistochemistry. Mouse brain sections were washed 3 times with PBS and permeabilized/blocked by incubating in 5% bovine serum albumin (BSA), 0.3% Triton X-100 in PBS for 1 h at room temperature. Incubation with primary antibodies, diluted in 1% BSA, 0.3% Triton X-100 in PBS, was performed overnight at 4 °C. The antibodies and their working concentrations are shown in the Table 2.

The mouse brain sections were washed 3 times with PBS and counterstained with Alexa Fluor-labeled secondary antibodies (dilution 1:400) for 2 h at room temperature. After washing 3 times with PBS, the mouse brain sections were treated with TrueBlack^®^ Lipofuscin Autofluorescence Quencher (Biotium, 23007, Fremont, CA, USA, dilution 1:20) for 1 min, and then, again washed 3 times with PBS. The slides were mounted with Prolong Gold Antifade mounting reagent with DAPI (Invitrogen, P36935) and analyzed using Leica DM 5500 Q upright confocal microscope (10×, 40×, and 63× oil objectives, N.A. 1.4). Images were processed and quantified using ImageJ 1.50i software (National Institutes of Health, Bethesda, MD, USA) in a blinded fashion. Panels were assembled with Adobe Photoshop.

### 2.8. Flow Cytometry Analysis of Dissociated Brain Cells and Splenocytes

The dissociated brain and spinal cord immune cells were obtained as described [23]. Briefly, an intact mouse spinal cord or the dorsal part of the left brain hemisphere, obtained from mice transcardially perfused with saline, were weighted, cut into very small pieces using a scalpel blade and digested using Collagenase D (Roche, Cat No. 11088858001, 1 mg/mL)/DNase I (Thermo Fisher, Cat No. 11875093, 50 µg/mL) solution in RPMI for 30 min at 37 °C. Cells were further purified, as described, by centrifugation in a 40–60% Percoll gradient in HBSS to remove debris and myelin [23]. Dissociated splenocytes were obtained from ~1/3 of the entire mouse spleen, as described [24], by forcing tissue pieces through a 100 μm nylon mesh cell strainer (BD Biosciences, Durham, NC, USA) submerged in ice-cold RPMI 1640. Splenocytes were collected by centrifugation at 300× *g* for 5 min at 4 °C.

The dissociated cells were labeled with anti-mouse CD45-BV785, CD11b-FITC, CX3CR1-BV605, CD68-BV421, CD63-Pe-Cy7, CD45.1-APC, CD45.2-Pe (all from Biolegend) and CD3-BUV395 (Becton Dickinson, Franklin Lakes, NJ, USA) antibodies. The cells were co-stained with the Aqua-Live/Dead fixable dye (LIVE/DEAD™ Fixable Aqua Dead Cell, Invitrogen Thermofisher) and fixed using 1% (*v*/*v*) PFA. Flow cytometry analysis was conducted using a Fortessa II instrument (Becton Dickinson) within 48 h following cell labeling and fixation. Data were analyzed using FlowJo X v10 and GraphPad Prism 9.5.0. software.

### 2.9. Analysis of Glycosaminoglycans by LC-MS/MS

Analysis of glycans in dry blood spots, plasma, urine, and brain tissues was conducted as previously described [9]. Briefly, 30–50 mg of mouse tissues were homogenized in ice-cold acetone and centrifuged at 12,000× *g* for 30 min at 4 °C. The pellets were dried, resuspended in 0.5 N NaOH and incubated for 2 h at 50 °C. Then, the pH of the samples was neutralized with 1 N HCl, and NaCl was added to the reaction mix at a final concentration of 3 M. After centrifugation at 10,000× *g* for 5 min at room temperature, the supernatants were collected and acidified using 1 N HCl. Following another centrifugation at 10,000× *g* for 5 min at room temperature, the supernatants were collected and neutralized with 1 N NaOH to a pH of 7.0. The samples were diluted at a ratio of 1:2 with 1.3% potassium acetate in absolute ethanol and centrifuged at 12,000× *g* and 4 °C for 30 min. The pellets were washed with ice-cold 80% ethanol, dried at room temperature, and dissolved in 50 mM Tris-HCl buffer. The samples were placed in AcroPrep^TM^ Advance 96-Well Filter Plates with Ultrafiltration Omega 10 K membrane filters (PALL Corporation, New York, NY, USA) and digested with chondroitinase B, heparitinase, and keratanase II, overnight at 37 °C. The samples were analyzed by mass spectrometry using a 6460 Triple Quad instrument (Agilent technologies, Santa Clara, CA, USA) with Hypercarb columns, as described [9].

### 2.10. Statistical Analysis

Statistical analyses were performed using Prism GraphPad 9.3.0. software (GraphPad Software San Diego, CA, USA). The normality for all data was verified using the D’Agostino and Pearson omnibus normality test. Significance of the difference was determined using *t*-test (normal distribution) or Mann–Whitney test, when comparing two groups. One-way ANOVA or Nested ANOVA tests, followed by Tukey or Dunnett multiple comparison tests (normal distribution), or Kruskal–Wallis test, followed by Dunn multiple comparisons test, were used when comparing more than two groups. Two-way ANOVA followed by Tukey post hoc test was used for two-factor analysis. A *p*-value of 0.05 or less was considered significant.

## 3. Results

### 3.1. Transplantation of MPS IIIC Mice with WT HSPC Results in a Partial Amelioration of the Disease Symptoms

To test whether allogeneic hematopoietic cell transplantation provides potential therapeutic benefits for MPS IIIC, we conducted a transplantation of HSPC from WT mice to *Hgsnat^P304L^* MPS IIIC mice. This strain, expressing a misfolded mutant HGSNAT enzyme, shows a faster disease progression compared to knockout *Hgsnat-Geo* mice, and recapitulates the clinical phenotype of early-onset human patients [8]. The study design is shown in Figure 1A. HSPC (CD34+) cells were isolated from the bone marrow of male B6.SJL-Ptprca Pepcb/BoyJ mice carrying the differential *Ptprc* pan leukocyte marker CD45.1 (Ly5.1) using EasySep Mouse Hematopoietic Progenitor Cell Isolation kit. Sixteen randomly selected male and female 8–9-week-old *Hgsnat^P304L^* mice (Appendix A) were transplanted with 2.4–2.8 × 10^6^ HSPC/mouse after a 5-day myeloablation procedure involving daily busulfan injections (IP 25 mg/kg BW/day). Two male mice died on the 4th and 21st days after transplantation. The exact cause of death was not determined, and we cannot exclude that it occurred due to complications related to the transplantation procedure. In addition, one female and one male mouse died 10 and 12 weeks after transplantation due to a malfunction of a water bottle and malocclusion, respectively. The other mice survived until the end of the experiment.

Six weeks after transplantation, the engraftment of HSPC was measured by flow cytometry analyses of CD45.1 expression in blood cell lineages. All mice showed engraftment between 82% and 94% (Appendix A). For the six untransplanted and nine transplanted *Hgsnat^P304L^* mice, as well as for the four untransplanted WT mice, dry blood spots (DBS) were obtained during the same blood collection procedure and used to analyze GAG levels by LC-MS/MS. This analysis determined drastically increased levels of HS-derived O-sulfated (ΔDiHS-OS) and N-sulfated (ΔDiHS-NS) disaccharides in the DBS of untreated *Hgsnat^P304L^* mice of the same age, while in the transplanted mice the levels of both disaccharides were not significantly different from the normal levels (Appendix A). This suggested that the transplantation completely or almost completely normalized HS levels in the plasma and blood cells of *Hgsnat^P304L^* mice.

Between the ages of 6 and 6.5 months (16 and 18 weeks post transplantation) the behavior of mice was analyzed using Open Field (OF, activity and anxiety), novel object recognition (NOR, short-term memory), contextual fear conditioning (CFC, associative learning) and Three-Chamber Sociability (TSC, general sociability and autistic-type behavior) tests. In the OF test, untreated *Hgsnat^P304L^* mice showed increased distance traveled in the field and number of entries to the central part of the arena (Figure 1B). These results, consistent with our previous data [8], are suggestive of hyperactivity and reduced anxiety, respectively. The total distance traveled for the transplanted group was similar to that of the WT group and statistically different from the untreated *Hgsnat^P304L^* group showing a rescue of hyperactivity. The transplanted mice also showed a non-significant trend towards reduction in the number of entries to the central area compared to the untreated group and were not statistically different from the WT group. The NOR test, with an interval of 1 h between encoding and retrieval, revealed that *Hgsnat^P304L^* mice spent much less time exploring the novel object than WT mice and showed a discrimination index close to zero (Figure 1C), consistent with impaired short-term memory. For transplanted *Hgsnat^P304L^* mice, both parameters, although showing a trend towards an increase, were not statistically different from those of untreated mice (Figure 1C). In CFC test, 2 and 7 days after the end of the fear-training period, untransplanted *Hgsnat^P304L^* mice showed significantly reduced freezing time revealing a problem of associating the environment with the fear-inducing stimulus (Figure 1D). Transplanted mice showed an increased freezing time compared to untreated *Hgsnat^P304L^* mice on day 7 of the experiment consistent with a partial recovery of associative learning, while no significant difference was observed on day 4. Finally, when mice were studied by the TCS test that assesses cognition in the form of general sociability, WT mice spent significantly more time in a chamber with access to a never-before-met intruder mouse than in the identical empty chamber showing a normal sociability typical for rodents (Figure 1E). In contrast, no statistically significant difference in the time spent in the chambers was detected for untreated *Hgsnat^P304L^* mice, which is consistent with MPS IIIC mice exhibiting deficits in social behavior and resembling autistic traits in human Sanfilippo patients. Transplanted mice showed a behavior similar to WT animals suggesting that they retained a normal sociability (Figure 1E).

Starting from ~7 months (30 weeks) of age, mice were evaluated daily by a veterinarian for signs of urinary retention (none, mild, moderate, severe, or very severe). Urinary retention is rarely seen in the patients, but it is developed by MPS III mice with all genetic subtypes [4,8,25]. This trait leads to distortion of the bladder and peritonitis and requires humane sacrifice of severely affected mice, thus limiting their lifespan. At the time of first observation, no mice in the transplanted group showed urinary retention, while one mouse in the untreated group showed mild and one mouse moderate signs (Figure 2A). The symptoms progressed with time and, by the age of 34 weeks, all untreated *Hgsnat^P304L^* mice developed urinary retention (one very severe, five severe, and five moderate) requiring their immediate sacrifice. In the treated group, only 1 of 11 mice of similar age had moderate signs and 5 mice had no signs of urinary retention (Figure 2A). Thus, allogeneic hematopoietic cell transplantation normalized or partially improved behavior deficits in MPS IIIC mice at the age of 6–6.5 months and delayed the onset of urinary retention, potentially increasing longevity.

All mice (both with and without urinary retention) were sacrificed by the age of 36 weeks (on average 28 weeks after the transplant) to analyze pathological changes in the CNS and peripheral tissues. At sacrifice, we measured the volume of urine retained in the bladder of mice. Untreated *Hgsnat^P304L^* mice had on average 1.5 mL of urine remaining in the bladder, while WT had none or less than 0.3 mL. The amount of urine in the bladders of transplanted *Hgsnat^P304L^* mice was on average equal to 0.5 mL and not significantly different from WT mice, consistent with the rescue of urinary retention (Figure 2B). We also measured the wet weight of the visceral organs (liver, kidney, and spleen) to assess the extent of organomegaly, reported for the *Hgsnat^P304L^* model [8]. Both treated and untreated *Hgsnat^P304L^* mice showed drastically increased relative weights of all three organs compared to WT mice (Figure 2C–E).

### 3.2. Transplantation Partially Increases HGSNAT Activity in Peripheral Tissues and Reduces Plasma Levels of HS

The levels of HGSNAT activity and total β-hexosaminidase activity were measured in the bone marrow, peripheral tissues (liver, spleen, lungs, and kidney) and brain of untransplanted and transplanted *Hgsnat^P304L^* mice. As expected, the HGSNAT activity was fully corrected in the bone marrow and the spleen. In the lungs and liver, the activity was increased on average to ~20% of normal, while in the kidney and the brain, HGSNAT activity showed only a minor increase and remained below 10% of normal (Figure 3A). These results indicate that transplantation of WT HSPC restored levels of the deficient enzyme in bone marrow and hematopoietic cells but not in other peripheral tissues or in the brain. In accordance with our previous findings [8], levels of total β-hexosaminidase activity, measured in the spleen, lungs, liver, kidney, and brain of untreated *Hgsnat^P304L^* mice, showed elevation compared to those in WT mice. In transplanted *Hgsnat^P304L^* mice, we observed a partial normalization of β-hexosaminidase activity in the lungs, liver, and the brain, but not in the lungs, spleen, or kidney (Figure 3B). In the bone marrow, total β-hexosaminidase activity was similar among WT, transplanted, and untransplanted mice (Figure 3B).

The results of HGSNAT and β-hexosaminidase assays were corroborated by the analysis of GAG levels in tissues, blood plasma, and urine by LC-MS/MS [8]. Levels of O-sulfated (ΔDiHS-OS) and N-sulfated (ΔDiHS-NS) disaccharides produced by enzymatic digestion of HS were drastically increased in all tissues as well as in the urine and plasma of untransplanted *Hgsnat^P304L^* compared to WT mice. In transplanted mice, levels of both disaccharides were reduced approximately two-fold in plasma but remained at levels similar to those of untreated group in urine, spleen, lungs, liver, kidney, and brain (Figure 3C,D). The levels of ΔDi-OS/4S (dermatan sulfate-derived disaccharide), as well as mono-sulfated (KS) and di-sulfated keratan sulfate (DiS-KS) were similar for untreated and treated *Hgsnat^P304L^* mice and their WT counterparts (not shown) with the exception of spleen, liver, and plasma, where KS was increased in untreated *Hgsnat^P304L^* mice and further normalized by treatment (Appendix A).

### 3.3. Transplantation of MPS IIIC Mice with WT HSPC Reduces Neuroinflammation

Microglia and tissue macrophages in the brain, spinal cord, and spleen of WT, *Hgsnat^P304L^* and transplanted *Hgsnat^P304L^* mice were analyzed by flow cytometry of dissociated cells to access their origin (host or transplant) and inflammatory phenotype. The gating strategy for dissociated brain/spinal cord cells and splenocytes is shown in Appendix A.

This analysis revealed that virtually all CD45/CD11b/CX3CR1-positive macrophages in the spleens of transplanted *Hgsnat^P304L^* mice showed CD45.1 phenotype, indicating that they were derived from transplanted HSPC (Appendix A). In both brain and the spinal cord, ~50% of CD45/CD11b/CX3CR1-positive macrophages/microglia cells were also CD45.1-positive (Appendix A) demonstrating that our experimental strategy resulted in effective population of the CNS by macrophages derived from the transplant.

To assess the inflammatory phenotype of macrophages/brain microglia, we measured fractions of cells expressing high levels of CD63 and CD68, robust markers of activation, in the total population of CX3CR1/CD11b-positive cells. In the spleens of WT mice, only 40% of CX3CR1/CD11b-positive cells expressed CD63 or CD68, while in the untreated *Hgsnat^P304L^* mice this number was above 60%. In contrast, in the spleens of transplanted *Hgsnat^P304L^* mice the percentage of CD63- or CD68-positive cells was not different from that in the WT mice indicating that the transplantation normalized activation of the resident splenic macrophages (Figure 4A–C).

Similarly, in the brain, the fractions of CD45/CD11b/CX3CR1-positive cells expressing elevated levels of CD63 and CD68 were <10% for WT mice, >80% for untreated *Hgsnat^P304L^* mice and ~50% for treated *Hgsnat^P304L^* mice (Figure 4D,E). Moreover, the treatment also normalized the total number of cells expressing elevated levels of CD63 or CD68 per mg of brain tissue that was drastically increased in the untreated *Hgsnat^P304L^* mice (Figure 4D,E). Consistently with the results of flow cytometry, the mRNA brain levels of pro-inflammatory cytokine IL-1β, expressed by activated microglia/macrophages, were drastically increased in the untreated *Hgsnat^P304L^* mice compared to the controls (Figure 4F). In the brains of transplanted *Hgsnat^P304L^* mice, IL-1β expression was reduced compared to the untreated *Hgsnat^P304L^* mice, although it remained elevated compared to WT controls (Figure 4F). In contrast, the expression levels of an anti-inflammatory cytokine TGF-β1 in the brains of transplanted mice were similar to those in untransplanted *Hgsnat^P304L^* mice (Figure 4G).

We further assessed the levels of CD63 and CD68 specifically in the fractions of CD45.1/CD11b/CX3CR1-positive and CD45.2/CD11b/CX3CR1-positive cells to estimate the activation levels of host microglia and those derived from the transplant (Figure 5A). CD45.1-positive microglia cells in the brains of transplanted *Hgsnat^P304L^* mice showed CD63 levels similar to those in the brains of WT mouse donors (Figure 5B). Moreover, the expression of CD63 in the CD45.2-positive (host) microglia was also significantly reduced compared to untreated *Hgsnat^P304L^* mice (Figure 5B). CD45.1-positive brain macrophages/microglia also showed a ~50% reduction in the CD68 levels, while CD45.2-positive cells showed a trend towards a reduction (Figure 5C).

Together, these data demonstrate that HSPC transplantation resulted in general reduction in neuroinflammation in MPSIIIC mice.

### 3.4. Transplantation Partially Normalizes the Levels of Biomarkers of CNS Pathology

The levels of activated CD68-positive microglia and GFAP-positive astrocytes, measured by immunofluorescent analysis of sagittal brain sections, were significantly increased in the hippocampi and somatosensory (layers 4–5) cortices of untreated *Hgsnat^P304L^* compared to WT mice (Figure 6A,B). The treatment reduced the levels of CD68-positive cells in both brain areas, while levels of GFAP-positive cells were normalized in the hippocampus only.

Previously, we have demonstrated that cortical pyramidal neurons accumulate autofluorescent materials that can be labeled with Thioflavin S and antibodies against amyloid-β protein [9]. The same cells also showed labeling with antibodies against misfolded subunit C of mitochondrial ATP synthase (SCMAS) suggestive of mitophagy block and a general impairment of proteolysis. When the levels of all four biomarkers were assessed in the cortices of untransplanted and transplanted *Hgsnat^P304L^* mice, we found that, while the levels of neurons showing autofluorescent inclusions, amyloid deposits, and SCMAS were ameliorated in the treated mice, they still remained considerably higher than in the WT cells, suggesting that transplantation could not completely correct this pathology (Figure 6C–E). We further analyzed levels of a G_M2_ ganglioside, known to be drastically increased in the brains of MPS IIIA-D patients [9] and mouse models [4,8], using the human–mouse chimeric monoclonal antibody, KM966 [26]. Numerous KM966-positive neurons were observed in the somatosensory cortex layers 4–5 of both treated and untreated *Hgsnat^P304L^* mice (Figure 6F). However, the amounts of G_M2_-positive cells showed a trend towards a decrease in the treated mice compared to the untreated mice. Consistent with the unchanged levels of the primary storage product HS and total β-hexosaminidase activity in the brains of treated mice, we found similarly increased size and abundance of LAMP-2 labeled puncta in NeuN-positive neurons in the brains of both treated and untreated *Hgsnat^P304L^* mice (Appendix A).

## 4. Discussion

MPS IIIC is one of multiple LSDs caused by defects in transmembrane proteins. Since such proteins are uncapable of a free diffusion between the cells and their cross-correction, it was assumed that heterologous HSPC transplantation or autologous HSPC gene therapy would not be effective for this disease. However, since the brain macrophages/microglia derived from the transplanted HSPC are capable of digestion of accumulating substrates, they are expected to develop a neuroprotective M2 phenotype and ameliorate the general levels of neuroinflammation in the recipient patients or animal models by secreting anti-inflammatory cytokines. In the current work, we tested this hypothesis by conducting a heterologous WT HSPC transplantation of MPS IIIC *Hgsnat^P304L^* mice with WT HSPC. Since HGSNAT is a multi-spanning transmembrane enzyme, we did not expect a neuronal cross-correction by donor-derived brain macrophages, so all improvements in CNS pathophysiology, if any, would result from the presence of macrophages with a “repair” M2 phenotype.

Our results established that heterologous transplantation of WT HSPC rescued urinary retention, the most severe life-threatening peripheral symptom in MPS IIIC mice. Unlike the untreated mice, transplanted *Hgsnat^P304L^* mice did not develop severe urinary retention by the age of 8 months, suggesting that the transplantation corrected inflammatory changes in the bladder walls and/or urethra, thought to be responsible for this trait [27]. Likewise, the levels of HS in the blood, measured 6 weeks after the transplantation, were completely normalized and remained 50% lower than those in the untreated mice until at least the age of 8 months. Additionally, in the spleen, liver, and plasma, we found a normalization of KS levels that were increased in untreated *Hgsnat^P304L^* mice. Potentially, KS could be increased due to a partial inhibition of KS-catabolizing enzyme, GALS by HS storage, but this requires further experimental confirmation. Surprisingly, the GAG levels in the liver, lungs, spleen, kidney, and urine remained similar to those in the untreated mice. For the spleen, these results were somewhat unexpected considering that the vast majority of spleen macrophages originated from the donor HSPC, and HGSNAT activity levels were normalized. It is tempting to speculate that HS level in the spleen reflects the accumulation of this substrate, that occurred before the transplant and was located in the areas not accessible for the infiltrating WT macrophages. Further studies are necessary to confirm this hypothesis. In other peripheral tissues, levels of HGSNAT activity showed a variable recovery to the levels from <3% of the WT (kidney) to 18–22% of the WT (lungs and liver). In the liver, we also observed a significant reduction in the levels of total β-hexosaminidase, known to be increased in the affected tissues of MPS III mouse models [4,8,25] but, like in the spleen, we did not observe any reduction in HS stored in these tissues.

Interestingly, patients with 15–20% residual enzymatic activity of normal present at the age of >60 years with retinitis pigmentosa instead of developing deadly juvenile phenotype like MPS IIIC patients with the residual activity of less than 3% [28]. Further studies are necessary to determine whether HSPC transplantation shows efficacy in improvement of the retinal phenotype present in MPS IIIC mice [29]. HSPC-derived cells were found in proximity to retinal epithelium in the transplanted mouse MPS models, suggesting a possibility of a potential amelioration of retinal pathology and improvement in retinal function [30,31]. However, the published studies reported the variable outcome in the transplanted patients affected with other forms of neurological mucopolysaccharidoses including those with MPS I (Hurler) and MPS III. In general, despite early improvement in ERG function, longer follow-up revealed a progressive decline in retinal function and ocular status [32,33]. Considering this, a local AAV-mediated gene therapy may become a better therapeutic option for such patients.

Our results also revealed an amelioration of some central manifestations of the disease in the transplanted mice. In the brain of transplanted mice, at least 50% of microglia/macrophages were derived from the donor HSPC, which was consistent with the numbers observed in other studies that used Busulfan conditioning [34,35]. At the same time, HGSNAT activity level in the brain, on average, remained below ~10% of the WT level, reflecting the contribution of the microglia to the total pool of this enzyme in the CNS. The absence of the enzyme deficiency correction was consistent with unchanged brain levels of HS, and along with LAMP-2 and total β-hexosaminidase activity, was recognized as one of the markers of lysosomal storage/lysosomal biogenesis. These results are highly consistent with those obtained for the mouse models of MPS IIIA and MPS IIIB, transplanted with WT HSPC [36,37]. Although both diseases are caused by defects in soluble enzymes, their levels of secretion from WT HSPC-derived brain macrophages were not sufficient for cross-correction of neurons or reduction in HS [36,37].

Other CNS disease biomarkers showed a diverse response to the therapy. The secondary storage materials, misfolded β-amyloid, SCMAS, and autofluorescent ceroid inclusions in the pyramidal cortical neurons, all associated with the autophagy block, were reduced while the levels of G_M2_ ganglioside remained unchanged reflecting the absence of cross-correction of neurons by macrophages. At the same time, levels of GFAP-positive astrocytes were drastically decreased in hippocampus, and the levels of CD68-positive activated microglia decreased in both cortex and hippocampus, which is suggestive of reduced neuroinflammation. This was further confirmed by a decrease in expression levels of proinflammatory cytokine IL-1β but not anti-inflammatory cytokine TGF-β1. Analysis of immune cells in the brain by flow cytometry provided an explanation for these observations: at least 50% of microglia in the brain of transplanted mice were CD45.1-positive and showed levels of activation markers similar to those in the brains of the WT mice. Moreover, activation of the resident CD45.2-positive macrophages was also reduced compared to the brains of untreated *Hgsnat^P304L^* mice, indicating that the donor-derived microglia could mitigate the neuroimmune response.

Notably, transplanted *Hgsnat^P304L^* mice showed a rescue of hyperactivity and reduced sociability, while the deficiency in associative learning showed only a partial improvement. These positive effects were in general not expected, since the overwhelming majority of neurons remained devoid of the enzyme. We speculate, therefore, that the observed effects are mainly due to the non-cell-autonomous effects of transplanted cells on neuroimmune response. This is also supported by published data showing that correction of neuroinflammation by hematopoietic stem cell gene therapy with IL-1Ra completely prevented the development of hyperactivity and spatial memory loss in MPS IIIA mice [38]. In contrast to this report, our data provide no evidence for a complete rescue of neuronal pathology and behavior deficits. One explanation for this could be that, in our study, HSPC transplantation was conducted at 2 months (8 weeks) of age. According to our published data [39], at this age, the pyramidal neurons of MPS IIIC mice already show synaptic deficits and significantly reduced density of dendritic spines. Similarly, levels of activated microglia show a drastic increase already at P25 [40]. Further studies are required to determine whether transplantation of mice at an earlier age would show efficacy in preventing CNS deficits; however, since most MPS IIIC patients are diagnosed post-symptomatically, we believe that our experimental design reflected the current patient situation. Also, it remains to be determined whether transplantation of gene-corrected autologous HSPC overexpressing supraphysiologic HGSNAT levels would correct the brain pathology as seen in the mouse models of MPS IIIA and MPS IIIB, both caused by deficiencies of soluble enzymes [37,41]. Lentivirus (LV)-mediated gene HSPC therapy was shown to improve or completely treat the disease in several animal models of neurological LSDs and is now approved for patients with metachromatic leukodystrophy [13,34,35,41,42,43]. Clinical trials for presymptomatic MPS IIIA patients are currently in progress (Manchester Hospital Clinical Trial 2023, Dr. Simon Johns, PI). Interestingly, HGSNAT has been detected in the extracellular vesicles (exosomes) isolated from culture medium conditioned with HEK 293T cells overexpressing the enzyme [44]. Thus, HGSNAT-loaded exosomes derived by microglia overexpressing the enzyme could potentially provide a certain level of neuronal cross-correction. Of note, our recent study [8] revealed that MPS IIIC mice treated by a HGSNAT chaperone, glucosamine, showed an improvement in CNS pathology, except for neuroinflammation that, as we show in the current work, is effectively treated by the HSPC transplant. It is tempting to speculate, therefore, that heterologous HSPC transplant used together with small-molecule drugs, either stabilizing a mutant HGSNAT enzyme and increasing its activity levels in the neurons (chaperone therapy) or reducing HS production (substrate reduction therapy), may be an effective component of a combination therapy for MPS IIIC.

## 5. Conclusions and Clinical Perspective

In this study, we tested whether neuroinflammation, a hallmark of the CNS pathology in neurological LSDs, can be ameliorated by brain macrophages/microglia derived from a heterologous transplant of healthy HSPC. We have selected for this purpose a mouse model of MPS IIIC, the disease caused by genetic deficiency of a transmembrane lysosomal enzyme HGSNAT, uncapable of free diffusion between the cells and their cross-correction. Together, our results demonstrate that the treatment improved microastroglyosis, expression of pro-inflammatory cytokine IL-1β, and delayed the development of terminal urinary retention, potentially increasing longevity consistent with the reduction in inflammatory response in the CNS and at the periphery. Treated animals also showed amelioration of several behavior deficits, including hyperactivity and reduction in socialization, suggesting that heterologous HSPC transplant may become an effective component of combination therapy for MPS IIIC in the future and, perhaps, for other LSDs caused by genetic defects in transmembrane proteins.

## Figures and Tables

**Figure 1 cells-13-00877-f001:**
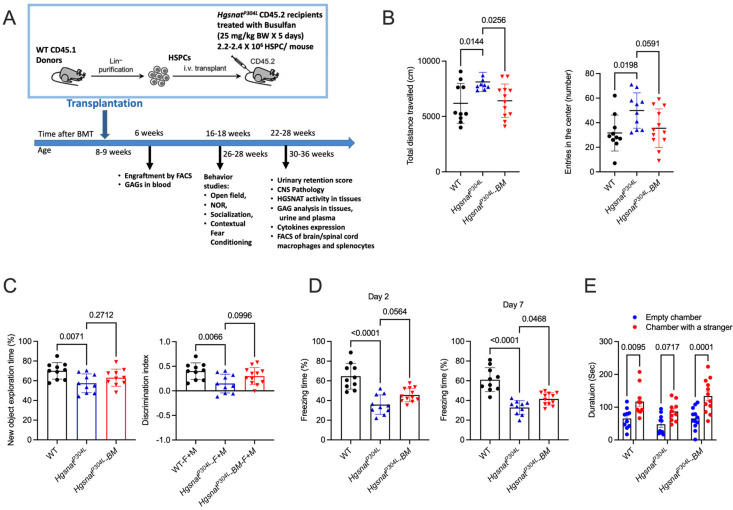
Transplantation of WT HSPC rescues several behavior abnormalities in *Hgsnat^P304L^* mice. (**A**) Experimental design. (**B**–**E**) WT, untreated, and transplanted *Hgsnat^P304L^* mice were analyzed at 6 months of age for presence of hyperactivity (increased total distance traveled) and reduced anxiety (increased number of entries to the center of arena) in the OF test (**B**), short-term memory deficit (reduced new object exploration time and discrimination index) using the NOR test (**C**), defects in associative learning (reduced freezing time) using the CFC test (**D**), and a reduced sociability (absence of interest in a stranger mouse) using the TCS test (**E**). Individual results, means and SD of experiments performed with 10–12 male and female mice per genotype, per treatment, are shown. (**A**–**D**) Mouse groups were compared with untreated *Hgsnat^P304L^* mice and *p* values calculated using one-way ANOVA with Dunnett post hoc test. (**E**) *p* values were calculated using two-way ANOVA with Tukey post hoc test.

**Figure 2 cells-13-00877-f002:**
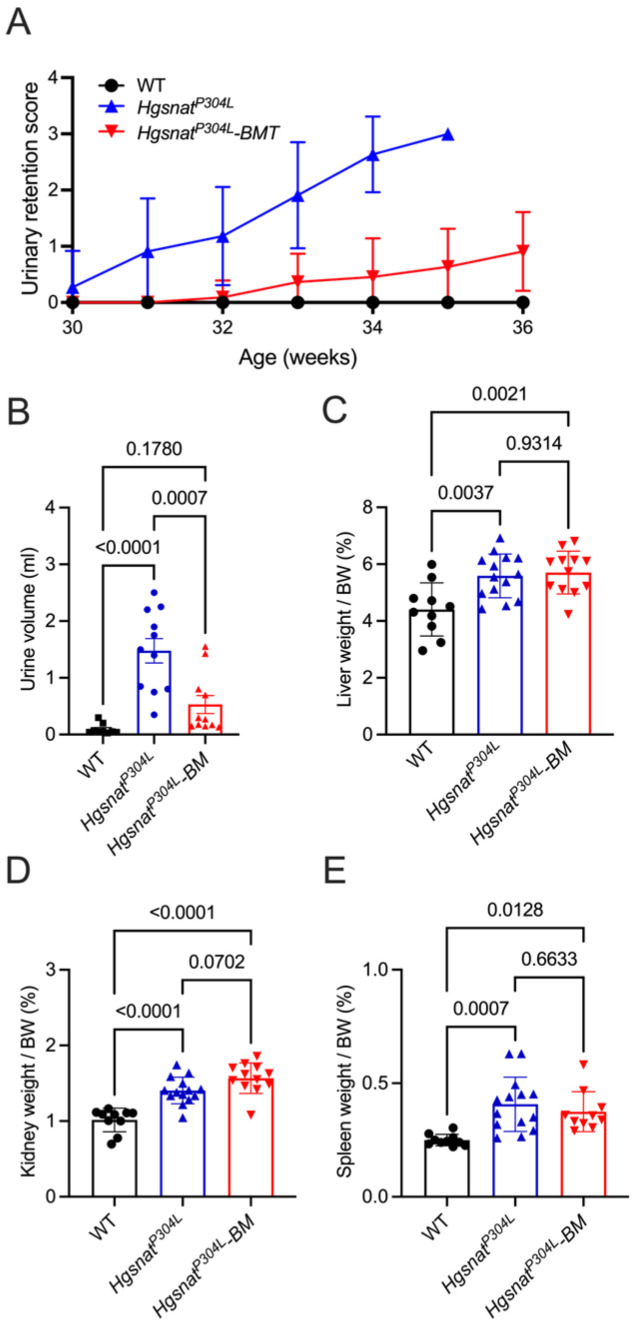
Transplantation of WT HSPC rescues urinary retention in *Hgsnat^P304L^* mice but does not alter organomegaly. (**A**,**B**) Development of urinary retention in untransplanted and transplanted *Hgsnat^P304L^* and WT mice. (**A**) Urinary retention score (0, none; 1, mild; 2, moderate; 3, severe; 4, very severe) was assessed for untransplanted and transplanted *Hgsnat^P304L^* male and female mice, and their WT counterparts (*n* = 16 for each group). The significance of differences between untreated and treated groups was determined by two-way ANOVA (*p* < 0.0001). By the age of 34 weeks, all untreated *Hgsnat^P304L^* mice but none of transplanted *Hgsnat^P304L^* mice had severe urinary retention requiring euthanasia. (**B**) Volume of urine in the bladder of mice at sacrifice. The amount of urine in the bladders of transplanted *Hgsnat^P304L^* mice is not significantly different from that in WT mice. (**C**–**E**) Wet organ weight of mice at sacrifice (34–36 weeks). Enlargement of liver (**C**), kidney (**D**), and spleen (**E**) compared to age-matched WT controls, consistent with the lysosomal storage and inflammatory cell infiltration, is detected in both untreated and transplanted *Hgsnat^P304L^* mice. Graphs show individual data, means and SD. *p* values were calculated using ANOVA with Tukey post hoc test.

**Figure 3 cells-13-00877-f003:**
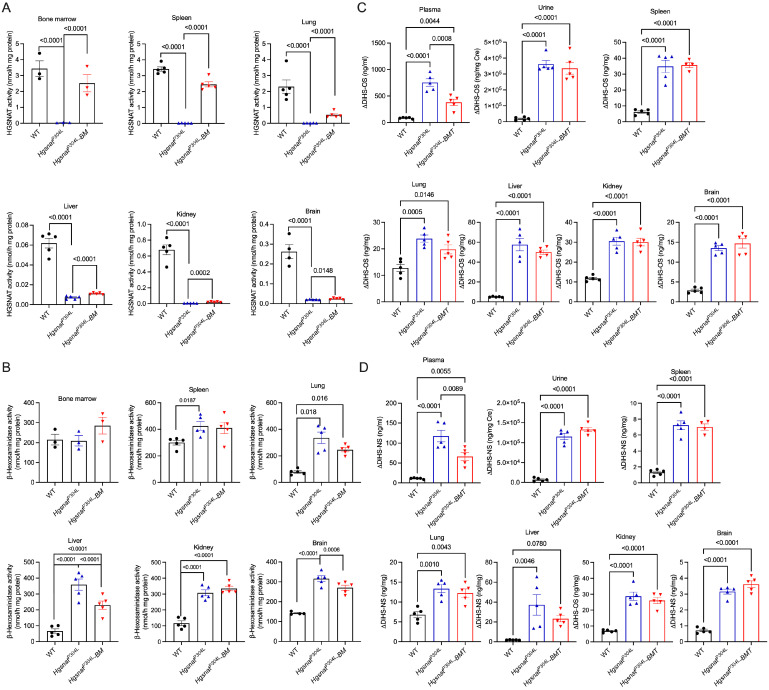
Transplanted *Hgsnat^P304L^* mice show normalized HS levels in plasma and partial rescue of HGSNAT deficiency and increased activity of lysosomal β-hexosaminidase in bone marrow and peripheral tissues, but not in the CNS. (**A**,**B**) HGSNAT activity (**A**) and total lysosomal β-hexosaminidase activity (**B**) in the tissues of 8-month-old WT, *Hgsnat^P304L^* and transplanted *Hgsnat^P304L^* mice. (**C**,**D**) Levels of disaccharides produced by enzymatic digestion of HS, ΔDiHS-OS (**C**), and ΔDiHS-NS (**D**) measured by tandem mass spectrometry in blood serum, urine, and tissues of WT, *Hgsnat^P304L^* and transplanted *Hgsnat^P304L^* mice. All graphs show individual data, means and SD of experiments performed using tissues from five male and female mice per genotype per treatment except for the bone marrow, where results for three samples, each containing pooled tissues of two mice, are shown. *p* values were calculated by one-way ANOVA with Tukey post hoc test. Only *p* values < 0.05 are shown.

**Figure 4 cells-13-00877-f004:**
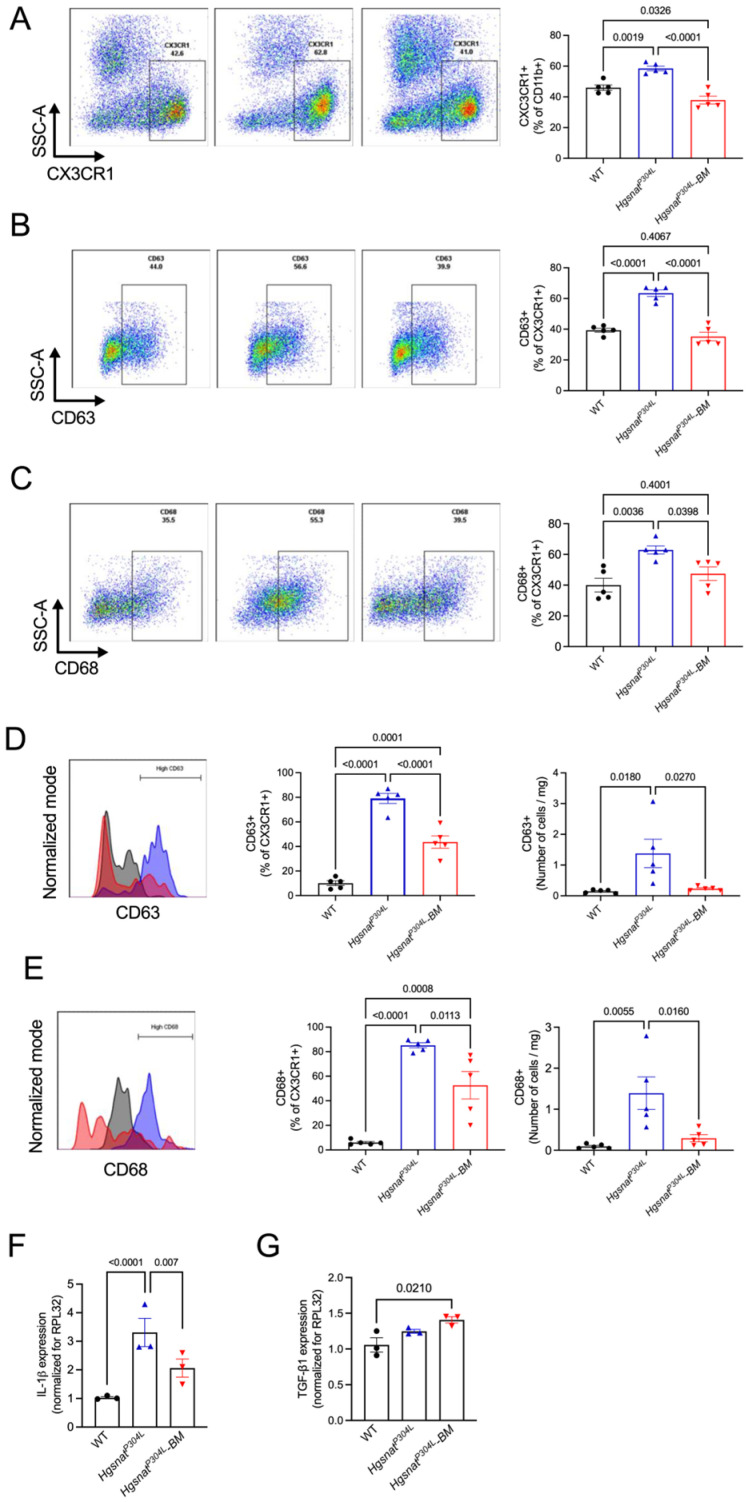
Systemic and brain inflammation is reduced in *Hgsnat^P304L^* mice transplanted with WT HSPC. To estimate inflammatory activation of spleen macrophages, frequency of CX3CR1 labeling on CD11b-positive cells (**A**) and CD63 (**B**) or CD68 (**C**) labeling on CX3CR1/CD11b-positive cells was evaluated in populations of dissociated splenocytes from WT, *Hgsnat^P304L^*, and *Hgsnat^P304L^* mice transplanted with WT HSPC (*Hgsnat^P304L^–BM*). To estimate the level of neuroinflammation, frequency for CD63 (**D**) and CD68 (**E**) was evaluated on CD45/CD11b/CX3CR1-positive cells in populations of dissociated brain cells of WT (grey traces), *Hgsnat^P304L^* (blue traces), and *Hgsnat^P304L^–BM* (red traces) mice. Data are shown as % of CD63+ or CD68+ cells among CD45+/CD11b+/CX3CR1+ cells (graphs on the left) or as total numbers of CD63+ or CD68+ cells per mg of brain tissue (graphs on the right). (**F**,**G**) Expression levels of pro-inflammatory cytokine IL-1β (**F**) were reduced in treated compared to untreated *Hgsnat^P304L^* mice while the expression levels of an anti-inflammatory cytokine, TGF-β1 (**G**), was unchanged. Graphs show individual data, means and SD for 5 mice per group (**A**–**E**) and 3 mice per group (**F**,**G**). *p* values were calculated using one-way ANOVA with Dunn post hoc test. Only *p* values < 0.05 are shown.

**Figure 5 cells-13-00877-f005:**
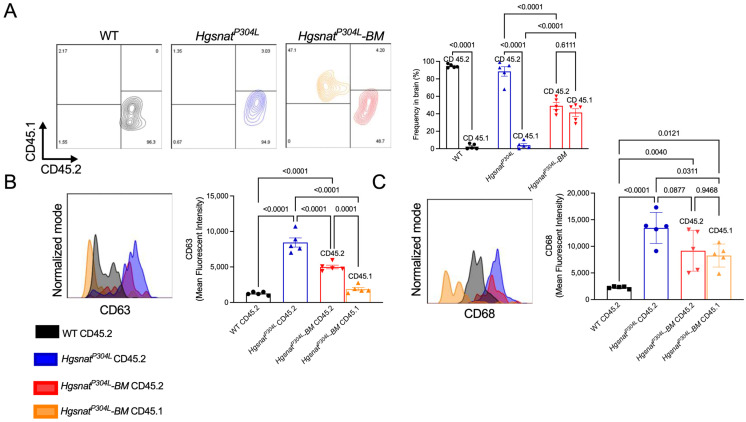
Reduction in neuroinflammation is mediated by infiltration of WT donor-derived CX3CR1/CD45.1-positive cells. (**A**) Frequency for CD45.1 and CD45.2 was evaluated on CD45/CD11b/CX3CR1-positive dissociated brain cells of WT, untransplanted *Hgsnat^P304L^*, and transplanted *Hgsnat^P304L^–BM* mice. (**B**,**C**) MFI for CD63 (**B**) and CD68 (**C**) was evaluated on CD45/CD11b/CX3CR1-positive dissociated brain cells of WT, *Hgsnat^P304L^*, and transplanted *Hgsnat^P304L^–BM* mice. Graphs show individual data, means and SD for five mice per group. *p* values were calculated using one-way ANOVA with Dunn post hoc test.

**Figure 6 cells-13-00877-f006:**
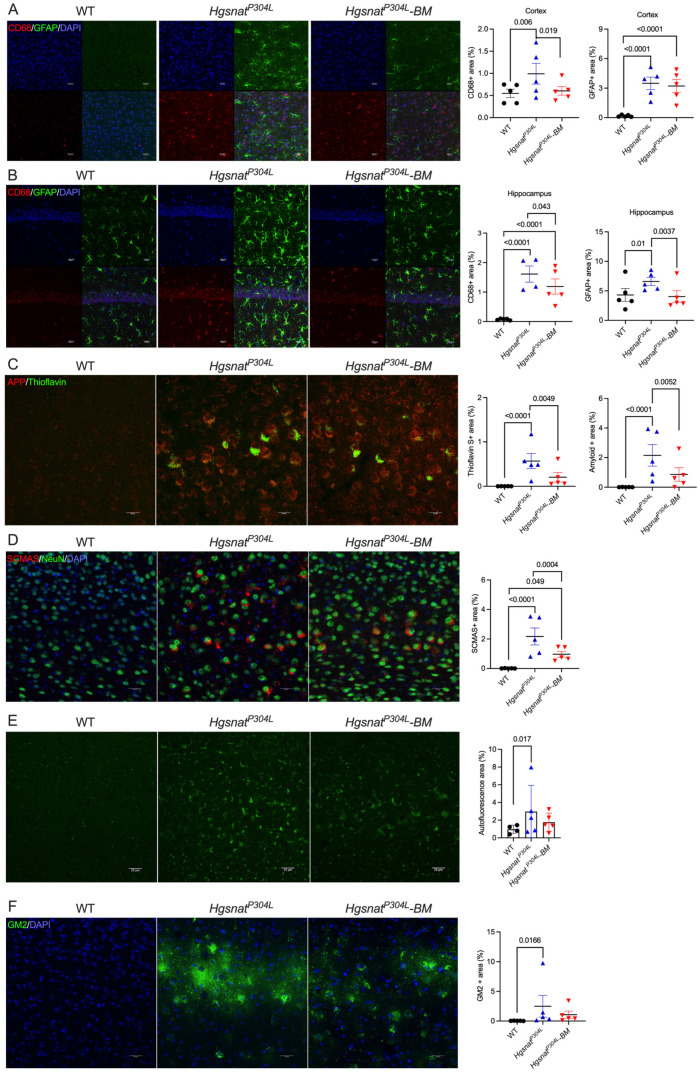
*Hgsnat^P304L^* mice transplanted with WT HSPC reveal recovery of astrogliosis and amelioration of levels of CNS pathology biomarkers in the somatosensory cortex and hippocampus. (**A**,**B**) Levels of activated CD68-positive microglia are reduced in the somatosensory cortices (**A**) and hippocampi (**B**) of 8-month-old transplanted mice compared to untransplanted *Hgsnat^P304L^* mice. Levels of GFAP-positive astrocytes are reduced in the hippocampus, but not in the cortex. Panels show representative images of the somatosensory cortex (layers 4–5) and the CA1 region of the hippocampus labeled for GFAP (green) and CD68 (red). DAPI was used as a nuclear counterstain. Bar graphs show quantification of CD68- and GFAP-positive areas with ImageJ 1.54i software. (**C**) Reduction in thioflavin S-positive β-amyloid aggregates in the cortical neurons of transplanted *Hgsnat^P304L^* mice. Panels show representative images of brain cortex (layers 4–5) labeled for β-amyloid (red) and misfolded proteins (Thioflavin-S, green). Graphs show quantification of β-amyloid and Thioflavin-S staining with ImageJ software. (**D**) Levels of misfolded SCMAS aggregates in cortical neurons are reduced in transplanted *Hgsnat^P304L^* mice. Panels show representative images of somatosensory cortex (layers 4–5), labeled for SCMAS (red) and neuronal marker NeuN (green). DAPI (blue) was used as a nuclear counterstain. The bar graph shows quantification of SCMAS staining with ImageJ software. (**E**) Reduction in granular autofluorescent ceroid material in cortical neurons. Panels display representative images of brain cortex showing autofluorescent ceroid inclusions in the neurons (green). The graph shows quantification of autofluorescence with ImageJ software. (**F**) Levels of G_M2_ ganglioside in cortical neurons are not reduced by transplantation. Panels show representative images of somatosensory cortex (layers 4–5) immunolabeled for G_M2_ ganglioside (green). The bar graphs show quantification of G_M2_ staining with ImageJ software. In all panels, scale bars equal 25 µm. All graphs show individual results, means and SD from experiments conducted with five mice (three panels per mouse) per genotype per treatment. *p* values were calculated using Nested one-way ANOVA test with Tukey post hoc test. Only *p* values < 0.05 are shown.

**Table 1 cells-13-00877-t001:** The forward (F) and reverse (R) primers used for qPCR experiments.

Gene	Sequence
F-IL-1β	TGAAATGCCACCTTTTGACA
R-IL-1β	GTAGCTGCCACAGCTTCTCC
F-TGF-β1	GTCAGACATTCGGGAAGCAG
R-TGF-β1	CTGCCGTACAACTCCAGTGA
F-RPL32	TTCTTCCTCGGCGCTGCCTACGA
R-RPL32	AACCTTCTCCGCACCCTGTTGTCA

**Table 2 cells-13-00877-t002:** The antibodies and their working concentrations used for immunochemistry.

Antigen	Host/Target Species	Dilution	Manufacturer
GFAP	Rabbit anti-mouse	1:300	DSHB, Iowa City, IA, USA (8-1E7-s)
Lysosomal-associated membrane protein 2 (LAMP-2)	Rat anti-mouse	1:200	DSHB (ABL-93-s)
NeuN	Rabbit anti-mouse	1:200	Millipore Sigma, St. Louis, MI, USA (MABN140)
CD68	Rabbit polyclonal to CD68	1:200	Abcam, Cambridge, UK (ab125212)
G_M2_ ganglioside	Mouse humanized	1:400	KM966 Kyowa Hakko Kirin Co., Ltd., Tokyo, Japan
β-Amyloid (D54D2)	Rabbit anti-mouse	1:200	Cell Signaling Technology Boston, MA, USA (8243S)
Recombinant Anti-ATP synthase C antibody (SCMAS)	Rabbit anti-mouse	1:200	Abcam (ab181243)
IgG	Goat anti-rabbit, anti-mouse, or anti-rat Alexa Fluor 488-, Alexa Fluor 555- or Alexa Fluor 633-conjugated	1:400	Thermo Fisher Scientific, Waltham, MA, USA

## Data Availability

The datasets used and/or analyzed during the current study are available from the corresponding author on reasonable request.

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
