# Peer review of "Heterologous HSPC Transplantation Rescues Neuroinflammation and Ameliorates Peripheral Manifestations in the Mouse Model of Lysosomal Transmembrane Enzyme Deficiency, MPS IIIC"

_cells, 2024, doi:10.3390/cells13100877_

Round 1
Reviewer 1 Report
Comments and Suggestions for Authors
The manuscript by Pan et al described a mouse model for HSCT in MPS IIIC. A novel and scientifically sound approach with a robust methodology and a succinct discussion. The data is potentially clinically relevant.
The authors could elaborate on a hypomorphic HGSNAT mutations leading to retinal phenotype only. How would the new potential therapy be effective (if at all) in treating patients with retinal phenotypes?
Author Response
We thank the reviewer for this important comment. We have revised the text of the Discussion section and added a description of an adult retinopathy caused by pathogenic variants in the HGSNAT gene that only partially reduce the enzyme activity. We have also included the following speculation on the potential impact of HSPC transplantation on retinal pathology (Lines 543-554).
“Interestingly, patients with 15-20% residual enzymatic activity of normal present at the age of > 60 years with retinitis pigmentosa instead of developing deadly juvenile phenotype like MPS IIIC patients with the residual activity of less than 3%[1]. Further studies are necessary to determine whether HSPC transplantation shows efficacy in improvement of the retinal phenotype present in MPS IIIC mice [2]. HSPC-derived cells were found in proximity to retinal epithelium in the transplanted mouse MPS models, suggesting a possibility of a potential amelioration of retinal pathology and improvement of retinal function [3,4]. However, the published studies reported the variable outcome in the transplanted patients affected with other forms of neurological mucopolysaccharidoses including those with MPS I (Hurler) and MPS III. In general, despite early improvement in ERG function, longer follow-up revealed a progressive decline of retinal function and ocular status [5,6]. Considering this, a local AAV-mediated gene therapy may become a better therapeutic option for such patients.”
Reviewer 2 Report
Comments and Suggestions for Authors
Pan et al studied the effects of heterologous HSPC transplantation on neuroinflammation and peripheral manifestations in a mouse model of MPS III. The study is very interesting because MPSIII is not due to deficiency of a lysosomal enzyme, that can be cross-correct, but a transmembrane enzyme. The study is well conducted, the results are clear and well described. The study could open the way to new therapeutic approaches.
I have only few suggestion about the introduction:
It is important to specify that neurological symptoms in MPS are due to heparan solfate storage, among GAGs (lines 48-49)
The neuroinflammation should be discussed before therapeutic approaches (lines 67-72)
Gene therapy is only mentioned but confusing given that the study is on HSTC transplantation
Author Response
Pan et al studied the effects of heterologous HSPC transplantation on neuroinflammation and peripheral manifestations in a mouse model of MPS III. The study is very interesting because MPSIII is not due to deficiency of a lysosomal enzyme, that can be cross-correct, but a transmembrane enzyme. The study is well conducted, the results are clear and well described. The study could open the way to new therapeutic approaches.
I have only few suggestion about the introduction:
It is important to specify that neurological symptoms in MPS are due to heparan solfate storage, among GAGs (lines 48-49)
As suggested by the reviewer, we emphasized that the neurological symptoms manifest only in the MPS patients accumulating heparan sulfate. “Neurological MPS (MPS I, MPS II, MPS IIIA, MPS IIIB, MPS IIIC, MPS IIID and MPS VII) are characterized by the accumulation of a specific GAG, heparan sulphate (HS) (reviewed in ref. [7]).” Lines 50-52.
The neuroinflammation should be discussed before therapeutic approaches (lines 67-72)
As suggested, we have added the description of neuroinflammation following the introduction of neurological forms of MPS. “HS, primarily stored in the phagocytic microglia and astrocytes, triggers their activation and release of pro-inflammatory cytokines and chemokines [3], which can further stimulate the upregulation of adhesion molecules on brain endothelial cells, resulting in enhanced trans-endothelial migration of monocytes from the blood into perivascular regions. These enzyme-deficient cells migrate into the parenchyma, leading to further damage and cell death (reviewed in ref. [2])”. Lines 52-57.
Gene therapy is only mentioned but confusing given that the study is on HSTC transplantation
We describe only one form of gene therapy – autologous transplantation of gene-corrected HSPC which is relevant to the subject of this study. To increase the clarity we have moved this description to the Discussion (lines 594-601).
Reviewer 3 Report
Comments and Suggestions for Authors
The manuscript by Pan and colleagues describes the application of an HSPC-based therapeutic approach on a previously chararacterized knock-in MPS IIIC mouse model. Authors performed an extensive characterization of behavioral, neuroinflammatory and biochemical changes produced in transplanted mice using a well-defined set of molecular, histopathological and enzymatic/biochemistry analyses.
I found in general the paper well-written and with a clear rationale. However, I noticed relevant misleading statements which currently prevent me from accepting the manuscript in its present form.
In particular, I realized that Authors improperly used many times the terms “non-significant trend” when pointing to slight but still non significant changes. This was already clear in Fig.1C when Authors claimed in the sentence at 291-292 lines that there was memory amelioration in transplanted mice when indeed there was no statistically significance between untransplanted and transplanted mice. Given this example as reference, I would encourage Authors to revise the manuscript according to the following points:
- Abstract (line 28-39). This part needs to be rephrased and all statements should be tuned down.
- Results (line 291-292). As stated above, this sentence is not correct. The lack of statistical significance points to the lack of memory deficit amelioration by transplantation.
- Results (Line 302-303). To be rephrased again. Similarity between control WT and transplanted mice could be inferred only by obtaining a significant difference between transplanted and untransplanted animals.
- Legend of Fig 1C. Adjust the text according to the changes requested before.
- Results. Lines 352 and 361. I personally do not agree with the statement that -hexosaminidase activity is per se a marker of lysosomal biogenesis. The proof of lysosomal biogenesis requires extensive molecular analyses which are far beyond the simplistic measurement of enzymatic activity for a single enzyme. I would encourage to remove this statement and rather assume that transplantation was able to partially recover HGSNAT activity in some organs.
- In Figure 3B the difference between -hexosaminidase in the lung is not statistically significant; therefore please correct line 362, accordingly.
- Line 374-376: This should be moved to the Discussion section. Indeed a conclusive sentence would be required to conclude this paragraph.
- Discussion section: as stated for the Abstract and Results sections, this part needs extensive rephrasing as data reported do not fully support Authors claims. As for example: “Our results established that heterologous transplantation of WT HSPC results in substantial amelioration of central manifestations”. This is obviously untrue. There are partial corrections at a behavioral, biochemical and molecular levels, in particular for target tissues and brain subregions (GFAP is not normalized in the cortex). Same correction needs to be done on lines 548-550, considering the partial normalization of behavioral parameters. The same applies for line 550.
- Minor changes: In the Fig S1 in the legend the "C" is missing"
Author Response
I found in general the paper well-written and with a clear rationale. However, I noticed relevant misleading statements which currently prevent me from accepting the manuscript in its present form.
In particular, I realized that Authors improperly used many times the terms “non-significant trend” when pointing to slight but still non-significant changes. This was already clear in Fig.1C when Authors claimed in the sentence at 291-292 lines that there was memory amelioration in transplanted mice when indeed there was no statistically significance between the untransplanted and transplanted mice. Given this example as reference, I would encourage Authors to revise the manuscript according to the following points:
We appreciate this comment of the reviewer. By the “non-significant trend,” we referred to the fact that Novel Object Recognition test (Figure 1C) revealed a significant difference (P<0.01) between the WT group and the group of non-transplanted HgsnatP304L mice in the time spent with a novel object and in the recognition index, while, for the treated mice, there was no difference with the WT group.
Saying this, we agree with the reviewer that this description could indeed be misleading for some readers, and we have revised the manuscript to provide the exact description of the experimental data and to make sure that no overinterpretation occurs. In all behaviour tests, the WT and transplanted HgsnatP304L groups were compared with the untreated HgsnatP304L group and the results were interpreted according to the significance of changes between transplanted and untreated HgsnatP304L groups.
Abstract (line 28-39). This part needs to be rephrased and all statements should be tuned down.
The abstract was edited as follows:
“We found that the treatment corrected several behaviour deficits, including hyperactivity and reduction of socialization, but not the memory decline. It also improved several features of CNS pathology such as microastroglyosis, expression of pro-inflammatory cytokine IL-1b, and accumulation of misfolded amyloid aggregates in cortical neurons.” Lines 29-32
Results (line 291-292). As stated above, this sentence is not correct. The lack of statistical significance points to the lack of memory deficit amelioration by transplantation.
This sentence was edited as follows: “For transplanted HgsnatP304L mice, both parameters, although showing a trend for increase, were not statistically different from those of untreated mice (Fig. 1C).” Lines 303-304
Results (Line 302-303). To be rephrased again. Similarity between control WT and transplanted mice could be inferred only by obtaining a significant difference between transplanted and untransplanted animals.
Does the reviewer refer to the lines 297-299, where we describe the results of the CFC (Figure 1D)? The interpretation of these data was changed to:” Transplanted mice showed an increased freezing time compared to untreated HgsnatP304L mice on the day 7 of the experiment consistent with a partial recovery of associative learning, while no significant difference was observed on the day 4.” Lines 307-309.
On lines 302-303 we described the results of sociability test, where for each group we were comparing the % of time spent interacting with the cage with the stranger mouse compared to an empty cage. We believe that these results were analysed correctly (presence of absence of a preference for a cage with a stranger mouse). WT and transplanted HgsnatP304L mice showed such preference and untreated HgsnatP304L mice, did not.
Legend of Fig 1C. Adjust the text according to the changes requested before.
The text of the legend was adjusted and all interpretations removed. Lines 318-326.
Results. Lines 352 and 361. I personally do not agree with the statement that b-hexosaminidase activity is per se a marker of lysosomal biogenesis. The proof of lysosomal biogenesis requires extensive molecular analyses which are far beyond the simplistic measurement of enzymatic activity for a single enzyme. I would encourage to remove this statement and rather assume that transplantation was able to partially recover HGSNAT activity in some organs.
We modified the text accordingly.
In Figure 3B the difference between b-hexosaminidase in the lung is not statistically significant; therefore please correct line 362, accordingly.
Corrected
Line 374-376: This should be moved to the Discussion section. Indeed a conclusive sentence would be required to conclude this paragraph.
This speculation was moved to Discussion. Lines 529-532.
Discussion section: as stated for the Abstract and Results sections, this part needs extensive rephrasing as data reported do not fully support Authors claims. As for example: “Our results established that heterologous transplantation of WT HSPC results in substantial amelioration of central manifestations”. This is obviously untrue.
This statement was changed to “Our results also revealed amelioration of some central manifestations of the disease in the transplanted mice.” and moved towards the second part of the discussion. Lines 555-556.
There are partial corrections at a behavioral, biochemical and molecular levels, in particular for target tissues and brain subregions (GFAP is not normalized in the cortex). Same correction needs to be done on lines 548-550, considering the partial normalization of behavioral parameters. The same applies for line 550.
These statements were corrected and now correlate with the observed significant changes. Lines 7579-580.
Minor changes: In the Fig S1 in the legend the "C" is missing"
Corrected
Round 2
Reviewer 3 Report
Comments and Suggestions for Authors
Authors have addressed most of my previously raised concerns, adjusting their overstating sentences and correctly considering my previous suggestions.
However, before accepting the manuscript in its final version I would again recommend Authors to reconsider their "statement" about beta-hexosaminidase as a lysosomal biogenesis marker, which I previously debated. There are still three places along the text where this arguable concept is mentioned. I am referring to the Figure 3 Legend and to Lines 539 and 559. Unless Authors can provide arguments about their statement I cannot accept the paper unless this correction is done.
Author Response
We appreciate the positive response of the reviewer regarding our previous revision. We would also like to provide a detailed response to the new critical comment below:
"However, before accepting the manuscript in its final version I would again recommend Authors to reconsider their "statement" about beta-hexosaminidase as a lysosomal biogenesis marker, which I previously debated. There are still three places along the text where this arguable concept is mentioned. I am referring to the Figure 3 Legend and to Lines 539 and 559. Unless Authors can provide arguments about their statement I cannot accept the paper unless this correction is done."
In the first review, the reviewer 3 wrote: “I personally do not agree with the statement that b-hexosaminidase activity is per se a marker of lysosomal biogenesis. The proof of lysosomal biogenesis requires extensive molecular analyses which are far beyond the simplistic measurement of enzymatic activity for a single enzyme. I would encourage to remove this statement and rather assume that transplantation was able to partially recover HGSNAT activity in some organs”.
This comment of the reviewer was interpreted by us as an optional suggestion to remove the statement that increased beta-hexosaminidase is a “marker of increased lysosomal biogenesis”. Accordingly, we removed this statement in the text of the results in the R1 revision. We now removed this statement also from the Figure 3 legend and edited it as follows: “Figure 3. Transplanted HgsnatP304L mice show normalized HS levels in plasma and partial rescue of HGSNAT deficiency and increased activity of lysosomal β-hexosaminidase in bone marrow and peripheral tissues, but not in the CNS.”
Sentence at lines 541-543 was changed to: “In the liver, we also observed a significant reduction in the levels of total b-hexosaminidase, known to be increased in the affected tissues of MPS III mouse models [4, 9, 26] but, like in the spleen, we did not observe any reduction of HS stored in these tissues.”
However in the discussion (Lines 561-563) we strongly prefer to use the statement edited as follows: “The absence of the enzyme deficiency correction was consistent with unchanged brain levels of HS and as well as, LAMP-2 and total beta-hexosaminidase activity, recognized as one of the markers of lysosomal storage/lysosomal biogenesis”.
In general, the concept of increased lysosomal biogenesis in the cells with lysosomal storage is well accepted (Settembre et al. Nature reviews 2013). It has been proven in many cases that the expression of the genes of the CLEAR network (such as HEXA and HEXB) is upregulated due to inhibition of TFEB phosphorylation at the lysosomal surface in response to the lysosomal storage. In particular, in our recent paper (Kho et al. JCI Ins. 2023, PMID: 37698928), we have demonstrated an increased translocation of TFEB to the nuclei and increased expression of multiple lysosomal enzymes and proteins in the kidney cortex of sialidosis mice. Increased levels of beta-hexosaminidase and other lysosomal hydrolases and proteins were also reported in the animal models of Sanfilippo diseases (see for example PMID: 10588735; PMID: 35704026; PMID: 25567323). The levels of beta-hexosaminidase activity, as well as levels of other lysosomal enzymes (such as beta-glucuronidase) or levels of lysosomal membrane proteins (LAMP-1/2 or LIMP II) measured by immunohistochemistry, were also used in multiple publications as a measure of a therapeutic response in the brain (PMID: 25267636; PMID: 33320673; PMID: 33230178; PMID: 15537895; PMID: 33320673). So by naming increased LAMP-2 and total beta-hexosaminidase activity “markers of lysosomal storage/increased lysosomal biogenesis” we do not claim that our results prove that the lysosomal biogenesis is increased in MPS IIIC mouse tissues but rather that these markers are pointing to such possibility.